# Food Allergy Education and Management in Early Learning and Childcare Centres: A Scoping Review on Current Practices and Gaps

**DOI:** 10.3390/children10071175

**Published:** 2023-07-06

**Authors:** Mae Jhelene L. Santos, Kaitlyn A. Merrill, Moshe Ben-Shoshan, Jennifer D. Gerdts, Don Giesbrecht, Elana Lavine, Susan Prentice, Julia Upton, Jennifer L. P. Protudjer

**Affiliations:** 1Department of Food and Human Nutritional Sciences, Faculty of Agriculture, University of Manitoba, Winnipeg, MB R3T 2N2, Canada; 2The Children’s Hospital Research Institute of Manitoba, Winnipeg, MB R3E 3P4, Canada; kmerrill@chrim.ca; 3Department of Biochemistry, Faculty of Science, University of Winnipeg, Winnipeg, MB R3B 2E9, Canada; 4Division of Pediatric Allergy and Clinical Immunology, Department of Pediatrics, McGill University Health Center, Montreal, QC H4A 3J1, Canada; moshebenshoshan@gmail.com; 5Food Allergy Canada, Toronto, ON M2J 4A2, Canada; jgerdts@foodallergycanada.ca; 6Canadian Child Care Federation, Ottawa, ON K1G 0Y9, Canada; 7Department of Paediatrics, Faculty of Medicine, University of Toronto, Toronto, ON M5G 1X8, Canada; 8Vaughn Pediatric Clinic, Woodbridge, Vaughan, ON L4L 8E2, Canada; 9Department of Sociology, Faculty of Arts, University of Manitoba, Winnipeg, MB R3T 2N2, Canada; 10Division of Immunology and Allergy, Department of Paediatrics, The Hospital for Sick Children, Toronto, ON M5G 1X8, Canada; 11Department of Pediatrics and Child Health, Rady Faculty of Health Sciences, University of Manitoba, Winnipeg, MB R3A 1S1, Canada; 12George and Fay Yee Centre for Healthcare Innovation, Winnipeg, MB R3E 0T6, Canada; 13Institute of Environmental Medicine, Karolinska Institutet, 171 77 Stockholm, Sweden

**Keywords:** anaphylaxis, early learning and childcare centres, food allergy, scoping review

## Abstract

Anaphylaxis has occurred in preschools/schools yet there are no consistent food allergy (FA) management practices in early learning and childcare centres (ELCC) across jurisdictions. Presently, there are no reviews that have synthesized FA-related knowledge and management practices within ELCC. We aimed to perform a scoping review of FA management in ELCC, and report on perceived gaps or barriers. A PRISMA-ScR-guided search was conducted for North American, European and Australian articles in English/French in the OVID-MedLine, Scopus, and PsycInfo databases. Two independent reviewers screened the titles/abstracts of 2010 articles and full-text screened 77 articles; 15 of which were specifically related to ELCC. If the two reviewers could not agree to the relevance of a given study, a third reviewer provided guidance. This third reviewer also screened French articles. Thematic and descriptive reports of the studies were presented. We reported solely on pre-Coronavirus Disease pandemic ELCC studies. We included ten articles in this review, which provide evidence that ELCC staff have variable baseline knowledge, comprehension, experience, and practices in place to manage FA. ELCC staff also have limited FA-related training and experience regarding administration of epinephrine auto-injectors (EAI). Emergency Anaphylaxis Plans (EAP) were described in four studies. One study reported the parental influence on the site’s food purchasing and FA management. Three studies provided educational interventions, which demonstrated increased and sustained FA-related knowledge and confidence post-intervention. Participants deemed the training beneficial and desired annual training and more FA resources to be available. Across jurisdictions, ELCC staff have provided care and administered EAI in emergencies, but training remained variable. Communication and care planning amongst ELCC staff, and parents, is crucial. Annual education, available EAI and EAPs are tools necessary for effectively managing emergencies.

## 1. Introduction

A food allergy, as defined by Boyce et al. (2010), is “an adverse health effect arising from a specific immune response that occurs reproducibly on exposure to a given food” (p. 11) [1]. Food allergies affect approximately 7.0–8.0% of children [2,3,4] and as many as 10.0% of infants and preschoolers (<5 years old) globally [5,6].

Food allergies are commonly diagnosed in infancy and the early childhood years [1,7,8]. An estimated 5–15% of all food allergic reactions occur in early learning and childcare centres (ELCCs) [9,10]. Moreover, it was reported that an estimated one-third of all food allergic reactions occurring at ELCCs were amongst children who reacted for the first time, or amongst children whom staff were unaware had a diagnosed food allergy [11,12]. These studies were completed approximately two decades ago, and more current estimates are lacking.

The first-line treatment for anaphylaxis, the most severe allergic reaction, is epinephrine, often administered through an auto-injector [1,13]. The availability of emergency anaphylaxis plans and access to prescribed and stock (i.e., unprescribed) epinephrine auto-injectors (EAIs) are essential in anaphylaxis management [1]. Yet, epinephrine remains underused in community settings, including ELCCs [14]. Suboptimal epinephrine use has been attributed to barriers affecting access, including unavailability [15], it being locked in an office/cabinet [10,16,17,18,19,20], administrative errors [15,21], caregiver failure to recognize symptoms [22], or fear/uncertainty about when to administer it [22].

Many families rely on childcare services to supervise their preschool-aged children outside of home. For example, about half (52.0%) of Canadian children <6 years old attend some form of ELCC, including unregulated home-based sites with non-relatives [23]. Preschool-aged children have higher supervisory needs than school-aged children, as reflected in child-to-adult ratios. Although older children (>13 years) may have a higher risk of allergic reactions from accidental exposures [24], younger children may be unable to voice symptoms to caregivers [20,22], and, in an emergency, need support in accessing and administering epinephrine [7,17]. Preschool-aged children with food allergies may also be undergoing oral immunotherapy or on partial-food-avoidance trials [25], which may require extra supervision at mealtimes.

Currently, no standardized food allergy or anaphylaxis preventative measures or management practices exist to guide Canadian or American ELCCs, despite available international recommendations [26,27]. For example, in Canada, management practices are often based on provincial or jurisdictional guidance, or implementation is specific to site-based discretion. ELCC facilities and programs may be owned privately or publicly and vary in site type (centre- or home-based), which may further impact the practices applied. In brief, food allergy training for ELCCs is inconsistently applied and delivered, and many centres lack access to EAIs unless provided by the parent/caregiver [15].

To the best of our knowledge, there are currently no reviews that have synthesized ELCC staff’s current food allergy knowledge of and management practices within ELCCs. In light of this knowledge gap, based on the above-cited findings, and through engagement with ELCC professionals, we aimed to perform a scoping review of food allergy management in ELCCs to identify perceived gaps or barriers in these practices.

## 2. Methods

Our scoping review was guided by the Preferred Reporting Items for Systematic reviews and Meta-Analyses extension for Scoping Reviews (PRISMA-ScR) 2020 Checklist [28]. A literature search of English and French articles in three medical literature databases (OVID-MedLine, Scopus, PsycInfo) was conducted for from the inception of each database to 19 February 2021. Schools and ELCCs (in the search strategy, early learning and childcare centres were termed “daycare” and “daycare centres” and “preschool”, as per recommendations from the expert librarian) were combined in our initial search strategy. Given the distinct differences as a result of developmental age, the disproportionately greater volume of data for food allergy management in ELCCs vs. schools, as well as the different policy and educational needs for ELCCs vs. schools, we decided to prepare stand-alone reviews. To enhance the study rigour and to update the original search, another search was run through PubMed on 21 November 2022. This additional search was performed to capture potentially relevant articles that used the term “early childhood education”. Herein, we operationalize ELCCs to childcare provided to children prior to the start of formal education (typically ages 5–6 years). Studies related to schools are reported elsewhere [29]. Studies with aggregate data on schools and ELCCs were also excluded later.

An expert librarian guided the search strategy used for the search. Each search was filtered to children (<18 years).

The inclusion criteria for which studies were screened were school and/or ELCC teacher or staff reports of food-allergy-related experience, education, and/or training in Canada, the United States of America (USA), Australia, and Europe (including Turkey). Articles that presented aggregate data on schools and ELCCs were also excluded to allow for analysis specific to school settings and ELCCs. Grey literature articles and abstracts and publications without original data were excluded in this review.

Our initial search, including both schools and ELCCs (Figure 1), found 2010 eligible articles (PsycInfo: *n* = 61; Scopus: *n* = 1414; OVID-MedLine: *n* = 535). De-duplication via Zotero removed 299 articles, and Rayyan software (Qatar Computer Research Institute, Doha, Qatar) [30] removed 10 articles. A total of 1701 articles were screened for titles and abstracts by two independent reviewers (initials blinded for review). No disagreements were found between screeners during the initial process. Full-text screening (*n* = 77) was completed for articles deemed eligible with consideration to the study methods, participants, outcomes of interest, and results. Of these 77 articles, 15 were deemed related to ELCCs. Two reviewers screened full-text English articles (*n* = 75), while French articles (*n* = 2) were screened by one reviewer.

Wrong outcome refers to a relevant paper that was identified through the search but was excluded because it did not consider the outcome of interest for our review.

## 3. Results

A total of ten articles were included in this review (Table 1)—two from Australia [18,31,32,33,34,35], and four from the USA [36,37,38,39], which we have presented thematically.

### 3.1. Experience Working with Children with Food Allergies

A wide range of food allergy experiences with preschool-aged children and the reporting thereof were noted amongst the studies. Australian, American, German, and Irish participants, across four studies, reported currently caring for children with one or multiple food allergies (9.0–83.0%) [31,32,35,36]. Meanwhile, 7.6% of Finnish children were on “allergy diets”; however, it was not reported whether all children on “allergy diets” had a food allergy diagnosis [33]. Overall, the estimated prevalence of food allergies for German and Irish children ranged from 1.5% [34] to 3.0% [35]. Another American study reported that 24.7% of participating ELCCs, whose facilities and programs catered to from <20 to >80 children per site, had >10 children with food allergies enrolled [37].

One study reported on cases of anaphylaxis amongst German children, 26.4% of which occurred in schools/preschools. Notably, the same study reported that ELCC staff had higher rates of reported anaphylaxis under their care compared to teachers in the same city (9.0% vs. 5.0%, respectively) [34], which may reflect that the children with food allergies had their first allergic reactions in ELCCs.

Information on nursing resources were limited. One study disclosed that school nurses were not involved in public German ELCCs [34], while participants in one American study included nurses; however, whether the nurses worked in ELCCs was not described [39].

#### Current Knowledge

Amongst ELCC staff, general food allergy and anaphylaxis knowledge varied widely. Participating ELCC staff had varied knowledge of diagnostic tests, common foods for infant feeding to reduce the risk of food allergies, potential anaphylaxis triggers, and food allergy prevalence [33,34]. In some countries, a high proportion of ELCCs were able to report food allergic reaction signs and symptoms. Most Finnish staff (~95.0%; 393/414) knew that post-prandial symptoms, which present after eating, such as rashes, vomiting, or difficulty breathing, may be signs of anaphylaxis [33]. Elsewhere, about half of ELCCs were able to describe the same. Amongst Irish ELCC staff, of whom a small proportion (16.0%; 15/98) reported on-site allergic reactions, >40.0% reported knowing when and how to use an EAI [35]. The following quote from an American ELCC provider reflects ELCC staff’s potentially inaccurate food allergy knowledge:

“*…Luckily, the only severe ones (food allergies) are the nuts, and so the peanuts, and so since we don’t allow them at all in the building, we don’t have to worry as much about that one*.”([38], p. 731)

Previous EAI or medication experience was similarly variable. Amongst German ELCC staff, 63.0% (47/75) were “familiar” with food allergies, 11.0% (8/75) of whom had previous experience in a food-allergy-related emergency, while 3.0% (2/75) had previously administered (unspecified) rescue medication [32]. Unspecified rescue medication was also provided to one child each by another group of German school teachers and early childcare educators [34]. Similarly, 4.0% (6/70) of American ELCC staff had previously administered EAIs [36]. At baseline, the same group scored a median of 3/5 (Standard Deviation (SD) = 1.3) for correctly sequencing the steps in administering EAIs (Foster et al., 2015 [36]). Elsewhere, one-quarter of Australian ELCC staff (26.4%; 14/53) had previous experience with food allergy emergencies [31].

### 3.2. Previous Training

Previous reports on food allergy training for early childcare educators and ELCC staff suggest that the training duration, content, and delivery are inconsistent. Inconsistencies persist, even within the same country. For example, a USA-based food allergy training program included a 45 min standardized presentation by a registered nurse who served as a food allergy educator, with hands-on opportunities with EAI trainers [39]. Another USA-based training program was guided by pediatric food allergy experts, in consultation with community members affected by food allergies, which lasted one hour and was delivered by a pediatric resident physician and pediatric food allergist [37]. Yet another USA-based study involved 40 min of content delivered by a physician, which was formally evaluated [36].

Given these differences in training, it is unsurprising that the proportions of early childcare educators and ELCC staff who reported receiving food allergy training also differed. A study conducted in the state of Washington, USA, reported that 42.9% (105/245) of ELCC staff had previous food allergy training, while 3.0% (3/105) of participants had previously administered epinephrine to a child in their care [39]. Comparatively, 43.0% (46/93) of ELCC staff based in Texas, USA, had previous training, primarily informal (e.g., self-taught, having received information from families or non-academic Internet resources) [37]. In this group, 35.4% (33/93) of participants had experienced an allergic reaction on-site in the last year, and 7.0% (6/86) had previously administered epinephrine. A third American study reported that more early childcare educators had training for EAI administration (70.0%; 120/171) than anaphylaxis recognition (57.3%; 98/171) [36].

Similarly varied are reports of Australian ELCC staff training. In one study, 91.0% of Australian ELCC staff from Western Australia reported that all caregivers were trained. Of these ELCCs, 7.0% (*n* = 3) did not require first-aid or anaphylaxis training, which the authors noted was “non-compliance” with Western Australian legislation, as at least one ELCC staff member trained in anaphylaxis recognition and management, including EAI administration, was required to be on-site [31]. One-third of food coordinators, who prepared food/menus, were also given training [31]. Training was primarily provided by first-aid professionals, while 22.0% (12/53) of ELCCs sought free online resources. Some reported barriers to training included costs and the availability of training sessions [31]. In another study, similar proportions of ELCC staff from Queensland, Victoria, and New South Wales had prior first-aid training, which included anaphylaxis training (90.5%; *n* = 447), while 6.7% (*n* = 33/494) required no training, and 31.8% (157/494) of respondents wanted in-person training [18]. “Other” types of training were wanted by 42 participants, but the study authors did not explain further. Interestingly, authors also reported on statistically significant differences in food allergy and anaphylaxis training amongst Australian jurisdictions. For example, more Victoria-based services required training on recognizing allergic reactions, including anaphylaxis, compared to services based in Queensland and New South Wales (87.3% vs. 75.0% vs. 66.1%, respectively; χ^2^(2, *n* = 441) = 17.24, *p* < 0.001) [18].

In an Irish study, the proportions of ELCC staff who had received food allergy and anaphylaxis training vs. those with no previous training was comparable at 19% [35]. In contrast, 3% were unsure whether they had received training, and data were missing for 58%. Of those who had received training, the majority of the training (60%) was delivered by health professionals [35]. The lack of training/resources was emphasized by a participant’s comment: “*Food allergy information needs to be more readily rolled out and not just sought when a child with a known allergy attends*” ([35], Para. 6).

### 3.3. Food Allergy Management

#### 3.3.1. Emergency Anaphylaxis Plans and Protocols

Food-allergy-related protocols or practices were reported in six (60.0%; 6/10) studies (Table 2). Reporting of emergency anaphylaxis plan availability was limited. One German study stated that children with food allergies had “emergency kits” with emergency anaphylaxis plans [34]. Written emergency plans were available in 47.0% of Irish facilities and programs (*n* = 46), but only half (55.0%, *n* = 24) had individualized emergency anaphylaxis plans created by ELCC staff or parents [35].

#### 3.3.2. Medications

##### EAI Availability and Other Medications

EAI availability was reported in six studies (6/10; 60.0%). Two studies reported having unprescribed, or stock, EAIs [18,31], while two studies did not specify whether the available EAIs were stock or prescribed [35,36]. In one German study, the EAI was prescribed as part of the “emergency kit”, while another German study reported that parents had to inform ELCC staff whether their child had an EAI and how to use it [33,34].

One or more medications were available in 32.0% of Irish ELCCs (*n* = 31): epinephrine (*n* = 24), inhalers (*n* = 13), and/or antihistamines (*n* = 15). Irish ELCC staff had medication-related uncertainties; authors quoted the participants’ free-text comments on drug administration and their desire for extra training [35]. Meanwhile, a German study reported that 71.0% of ELCC facilities and programs (43/61) had unspecified “rescue medication” [32].

##### Medication Knowledge

Within Western Australian facilities and programs, 96.0% (51/53) responded that ELCC staff knew the location of the EAIs, 36.0% (18/51) of which were locked in offices. Most ELCCs (68.0%; 36/53) had stock EAIs, while 31.0% (11/36) had a range of 2–8 EAIs available per site. In sites without stock EAIs (32.0%, *n* = 17), 16 ELCCs had at least one child at risk for anaphylaxis. Some sites also had EAIs in field-trip bags (9.0%; 5/53) and EAI training devices (58.5%; 31/35) [31]. Similarly, among Australian ELCC staff from Queensland, Victoria, and New South Wales, 97.3% knew the location of the EAIs, but only 95.7% had access. Approximately one-third of participants (37.0%) reported that their EAIs were stored in a locked location. At least one stock EAI was available on-site according to 47.6% of Australian participants [18].

In contrast, 11.4% (15/132) of American ELCC staff, including directors, teachers, assistants, and other staff, were uncertain of the EAI location. Some ELCC staff (4.0%; 6/170) had previously administered EAIs, although, concerningly, 19.0% (14/73) of early childcare educators were unauthorized to administer EAIs. Whether staff who had previously administered EAIs were all permitted was unspecified [36]. American ELCC staff reported parents were almost always called about on-site food allergic reactions (97.0%; 32/33), where EAIs were administered in 9.0% (3/33) of cases and antihistamines were given more times (51.5% 17/33) [37].

Further, 60.9% (53/87) of German kindergarten and primary school children who previously experienced anaphylaxis were prescribed “emergency kits”. Kits contained an emergency anaphylaxis plan, medication, and instructions. Kits were deemed “correct” if an EAI was included. Medication had varied prescribers (pediatrician, allergist, general physician) and different forms (EAI, corticosteroids, B2 antagonists, and antihistamines). More ELCC staff compared to teachers knew their students’ emergency kit contents (87.0% vs. 66.0%, respectively, *p* = 0.05) and had used the kit more (49.8% vs. 11.1%, respectively, *p* = 0.05).

##### Confidence

Regarding food allergy management-related confidence, all Western Australian early childcare educators and ELCC staff reported being “confident” or “very confident”, as their services always have food-allergy-trained staff for EAI administration and food provision. Nevertheless, 6.0% (3/53) of Western Australian ELCC staff were concerned about allergic reaction recognition and individual roles in an emergency [31]. In another Australian study of ELCC staff from Queensland, Victoria, and New South Wales, there was no significant difference between staff who had related training compared to those who did not have related training when asked about their confidence related to emergency medication administration, recognition of signs and symptoms of an allergic reaction, and role in an emergency response. However, participants who had the required training had significantly higher confidence levels in “providing appropriate/ safe food for a child with food allergy” compared to those who did not have the required training (χ^2^(2, *n* = 486) = 6.05, *p* = 0.048) [18].

A small proportion of German early childcare educators (11.0%; 8/75) reported being well prepared for emergencies [32]. Elsewhere, contrasting reports of uncertainty were reported amongst American early childcare educators and ELCC staff about anaphylaxis recognition. Over half of one group reported being uncertain of recognizing symptoms (69.0%; 91/132) and using an EAI (53.8%; 71/132) [36]. The same group reported comfort levels with anaphylaxis recognition and EAI administration pre-intervention at 5.1/10 (SD = 2.4) and 5.4/10 (SD = 2.8), respectively. Notably, significantly higher pre-intervention comfort levels were reported for ELCC staff with previous anaphylaxis recognition training than those without training (mean = 5.7/10 vs. 4.3/10, respectively; *p* < 0.001) and EAI administration training (mean = 5.9 vs. 4.0/10, respectively; *p* < 0.001) [36]. Conversely, more Texas-based ELCC staff self-reported higher/moderately higher abilities in recognizing symptoms (78.0%; 71/91) than treating anaphylaxis (61.1%; 55/90) and using an EAI (64.4%; 58/90) [37].

##### Barriers to Food Allergy Management

Perceived barriers in food allergy management were reported. American ELCC owners, program directors, and staff described food substitutions to accommodate allergies as requiring “considerable amounts of time” and costing more than other menu items. Similarly, from short-answer questions, Australian early childcare educators and ELCC staff were concerned about parents’ “apathy” in bringing in foods to which other children were allergic. Miscommunication or undocumented food allergies were other barriers hindering on-site management [31].

### 3.4. Food Provision and Preparation

Food allergy-related food preparation and meal service were discussed in two of the nine studies [33,38].

One USA-based qualitative study interviewed 16 ELCC representatives (*n* = 14 owners and program directors, *n* = 2 food service staff) regarding food-purchasing decisions for centre-based (*n* = 14) or home-based (*n* = 2) sites. Half of the ELCC facilities (7/14) had at least one employee tasked with ordering, preparing, and cooking. Menu planning was driven by staff’s personal preferences and available state licencing and federal healthy menu subsidy programs available for American ELCC services [38].

Food allergies were one of four themes described from interviews. Authors reported that “*almost all providers emphasized the high prevalence of child food allergies and intolerances—with some mentioning these were often undiagnosed*” ([38], p. 731). However, information about how allergies were deemed “undiagnosed” was not provided, nor whether or how these were managed differently than children with diagnosed food allergies. One participant discussed having a nut restriction on-site, although “…*some parents forget and bring things in*” [38].

Photographs illustrating the management of food allergies and other medical dietary restrictions were provided. Photographs showed home/commercial kitchen spaces, food storage areas, and detailed lists of children and their dietary restrictions. To accommodate children with food allergies and/or dietary restrictions, some parents brought in their own food substitutes, whereas some staff had to find alternatives, described as “expensive” and “time-consuming” [38]. “Many” providers also reported having family-style meals to encourage peer influence on children’s food choices; however, the approach to prevent children from sharing food, a critical element of food allergy management, was unexplained [38].

The other study, which described meal provision practices amongst ELCCs, was a Finnish longitudinal intervention study including 40 ELCC facilities and programs from three cities as part of the Finnish Allergy Program. Finnish ELCCs provided children with three meals from central kitchens (75.0%; 30/40) or made on-site (25.0%; 10/40). Pre-intervention, sites used different forms in order to remove or “avoid” up to 37–86 foods from the menu. A standardized “special diet form” was implemented as an intervention to reduce non-essential allergies, such as non-physician-diagnosed or parent-reported allergies. ELCCs who had not adopted the form yet were considered controls. Additionally, the new form required parents of children at risk for anaphylaxis to inform ELCC staff on how to use an EAI [33].

At baseline, 7.6% of children in 42 ELCCs (244/3216) were on allergy diets that primarily eliminated cow’s milk, eggs, and grains. Medical certificates were provided for only half (48.0%; 118/244) of the children on special diets, but not all were physician-provided, although signatories on non-physician-signed certificates were not reported [33].

The number of children with allergy diets decreased in the two-year study period, from baseline to the 2014 and 2015 follow-ups (7.6%, 244/3216; 7.3%, 236/3233; 4.3%, 148/3411, respectively; *p* < 0.001). The total number of avoided foods decreased amongst both the intervention and control sites during the study period (range = 1–33 and 1–53 foods, respectively; *p* = 0.107), while the mean number of avoided foods per child remained (median = 2, range = 1–63 foods). The rate ratio for the new special-diet-form use to the number of avoided foods per child was 1.59 after adjusting for city and ELCC site (95% Confidence Interval (CI) = 0.91–2.78; *p* = 0.10) [33].

Study authors reported positive responses from ELCC program directors and kitchen staff to the new special diet form. Half (52.0%; 35/67) of the respondents deemed the new practice “better” compared to the old form, and most (90.0%; 60/64) agreed that the new form was informative enough to provide children correct meals. However, reported barriers included inadequate training for new staff, lack of communication, and “uncertainty in commonly agreed practices”, although the latter was not further described [33].

### 3.5. Educational Interventions

Educational interventions were provided in German and American studies (44.4%; 4/9). All studies that provided interventions reported better food allergy and anaphylaxis knowledge compared to baseline (see Table 2) [32,36,37,39].

Texas-based early childcare educators had statistically significant questionnaire scores, from pre- to post-intervention, on understanding food allergies (62.6% vs. 85.2%, respectively; *p* < 0.001), reaction recognition (62.6% vs. 81.1%, respectively; *p* < 0.001), and laws on food labelling (40.0% vs. 76.3%, respectively, *p* < 0.001). Increased knowledge on treatment was also demonstrated from pre- to post-intervention, albeit non-significant (51.1% vs. 80.6%, respectively; *p* = 0.001) [37].

Amongst German early childcare educators, knowledge increased after a one-hour education session on food allergies and anaphylaxis, including an EAI demonstration. Knowledge on possible anaphylactic symptoms provided by the authors (fecal/urinary incontinence; dip in blood pressure and dizziness; shortness of breath, wheezing; nausea and vomiting; and swelling of the skin and mucosa) was better amongst participants from pre- to post-intervention and at follow-up (9.0% vs. 60.0% vs. 31.0%, respectively; *p* < 0.025). Similarly, ELCC staff recognized peanuts as a greater potential anaphylaxis trigger compared to other foods, such as eggs, cow’s milk, or apples. This proportion increased from pre- to post-intervention and at follow-up (87.0% vs. 96.0% vs. 87.0%, respectively; *p* < 0.025) [32].

Study authors reported that if participants were given “specific conditions like medical prescription, authorization or instructions”, then fewer German ELCC staff would administer an EAI in an emergency at baseline, post-intervention, and follow-up (47.0% vs. 33.0% vs. 31.0%, respectively; *p* < 0.025). However, those who would administer an EAI, even if no specific instructions were available, increased in the same time period (40.0% vs. 59.0% vs. 61.0%, respectively; *p* < 0.025) [32]. Although these results were contradictory among the same group of ELCC staff, the authors did not elaborate further.

EAI administration skills were also observed. Among German ELCC staff, a decrease in high-clinical-risk problems during EAI administration was noted. The most improvement seen from pre- to post-intervention and at follow-up was in unlocking (from 11.0% to 1.0% to 5.0%, respectively; *p* < 0.025) and activating (from 43.0% to 5.0% to 4.0%, respectively; *p* < 0.025) the EAI pre-administration. However, increased error was reported with the EAI administrator injecting their own finger (from 8.0% to 8.0% to 12.0%, respectively; *p* < 0.025) [32]. Similarly, American ELCC staff were able to order all five steps on proper EAI administration from pre- to post-intervention (15.0% vs. 57.0%, respectively; *p* = 0.04), which the study authors reported as “weakly but positively correlated with comfort level of EAI administration” [36].

Interventions also increased staff confidence. More German ELCC staff felt prepared for an anaphylactic emergency from pre- to post-intervention and at follow-up (11.0% vs. 88.0% vs. 79.0%, respectively; *p* < 0.001) and reported increased confidence in EAI administration within the same time period (median = 5/10 vs. 9/10 vs. 8/10, respectively; *p* < 0.001; on a scale of 1–10, with 10 indicating the highest confidence) [32]. Similarly ranked on a 1–10 scale (with 10 indicating the highest confidence), American early childcare educators and ELCC staff reported increased comfort levels from pre- to post-intervention in anaphylaxis recognition (5.1/10 vs. 8.7/10, respectively; *p* < 0.001) and EAI administration (5.4/10 vs. 8.8/10, respectively; *p* < 0.001). Notably, early childcare educators and ELCC staff with previous food-allergy-related training had significantly higher comfort levels with anaphylaxis recognition (mean = 5.7/10 vs. 4.3/10; *p* < 0.001) and EAI administration (mean = 5.9/10 vs. 4.0/10; *p* < 0.001) compared with those without previous food allergy training [36].

In another USA-based cohort, 94.0% of participants, of whom 6.0% (246/4088) were early childcare educators, reported increased confidence post-intervention and at 3–12-months follow-up. Post-intervention, 62.4% (153/245) of early childcare educators reported a higher likelihood of changing current food allergy management practices [39]. At follow-up, over half of participants (57.0%; 188/332) were able to recall three key messages from the intervention. Additionally, 21 participants from the same study, 1 of whom was an early childcare educator (others were teachers, program administrators, and other staff), reported partaking in a food allergy emergency since the intervention. Half of these emergencies (57.1%; 12/21) occurred in ELCCs, and 42.9% (9/21) were caused by previously unknown food allergens [39].

### 3.6. Future Food Allergy Training

Amongst American school teachers, ELCC staff, and community workers who had responded in actual food allergy emergencies post-educational intervention, approximately half reported that the session (62.0%; 13/21) and EAI administration training (52.4%; 11/21) were useful in responding to the emergencies [39]. Similarly, 99.0% of American participants believed the educational intervention was beneficial, and 76.0% wanted annual training. Specifically, 90.0% reported wanting more training in anaphylaxis education [36].

Survey open-ended questions revealed that Australian ELCC staff and Irish staff were interested in more food allergy and EAI training [31,35]. Some comments related to future training needs included having more “detailed” and hands-on training, including refresher courses [31,35]. Australian participants also wanted affordable and after-hours training, and they believed sharing food-allergy-related resources may increase parent communication [31].

## 4. Discussion

In this scoping review, we provide a descriptive summary of food-allergy-related management practices in ELCC services. Perceived gaps and barriers to food allergy management included inconsistent knowledge, training, and epinephrine availability and administration. Our review emphasizes the unique ELCC environment and the interdependent responsibilities of early childcare educators, other childcare staff, and parents. Given that most preschool children are pre-literate and, hence, may be unable to read food labels, and given that they lack language skills to fully articulate symptoms, they are at increased risk of accidental exposure to their allergens.

Thus, the need to provide the ELCC sector with adequate support, training, and resources to prevent and manage food allergy emergencies is warranted and supports recent food allergy guidelines and recommendations [27].

A minority of studies reported on parents’ roles in food allergy management, which impacted staff’s management strategies, including unintentionally bringing in restricted foods and driving menu-planning decisions [31,38]. Jacobsen et al. [31] also noted that ELCC workers reported that enhanced communication with parents of children with food allergies and without food allergies may help address barriers that persist despite existing food allergy education resources. The ELCC size also varies, wherein smaller sites may be unable to divide children based on the levels of care needed [17]. As such, specific food restrictions, educational resources, and communication provided to all parents are also important to protect children with food allergies.

Additionally, we reported on American ELCC staff, including early childcare educators who were not authorized to administer epinephrine [36], or would not provide treatment without “specific instructions” [32] in an emergency. This is concerning, as most ELCC staff are likely the first adults to witness and/or attend to an allergic reaction for a child in their care.

Two studies provided evidence that meal-planning practices were inconsistent in managing and organizing diets for children with food allergies [33,38]. Participant quotes related to food provision also suggested that ELCC staff’s personal beliefs, food preferences, and potentially inaccurate food allergy knowledge may impact meal provision practices. Thus, mandatory safe food handling and food allergy training may hold value for all ELCC staff. As a result of cultural, policy, and legislative differences between jurisdictions, as well as food supply chains, financing structures, and available funding, some ELCC centres may opt to include pre-packaged foods; in this light, training on label reading is also a necessary skill for ELCC staff. Additionally, following the intervention, the number of children following allergy-restrictive diets decreased from 7.6% to 4.3%, corresponding to a decrease of 43% [33], which suggests that a standardized food allergy form may benefit facilities to efficiently organize and provide safe meals for all children. The positive acceptance of educational interventions and the desire for more training [31,36,39] also emphasize the perceived need and value by early childcare educators and ELCC staff.

Lastly, studies in this review that provided educational interventions reported similar results in increasing food allergy and anaphylaxis knowledge to those of studies previously published [15,40]. Where food allergy training is already available, knowledge assessment through written evaluation and EAI administration demonstration, as well as incentivizing ongoing training by providing continuing education credits [39], may be considered by jurisdictions to retain food allergy knowledge and boost staff confidence related to emergency management. Moreover, we reported variable proportions of ELCCs with available EAIs and emergency anaphylaxis plans. In line with current international recommendations [27] and the possibility that preschool-aged children are on oral immunotherapy or may follow partial food avoidance [25], programs and facilities should consider adapting guidelines and implementing policies on reducing the risk of reactions and provide staff with resources to effectively and safely manage emergencies, such as having up-to-date emergency anaphylaxis plans and available EAIs, as well as safe mealtime practices, such as avoiding food sharing and promoting handwashing.

This is the first scoping review to report an overview of the current practices and knowledge of ELCC staff and providers amongst international jurisdictions. We searched three medical literature databases to ensure a broad literature search, and we included studies in multiple languages. Our review highlights the screening and data extraction work of two researchers, while all content was subjected to review by all other collaborators. Our review also reports on the positive uptake and benefits of food allergy and anaphylaxis training, including the administration of EAIs, while noting the related knowledge and gaps in current practices. We acknowledge that, given the structure of the study, a quality appraisal of the included studies, comparisons of interventions, and provision of a cohesive analysis of the study results were limited [41]. Information and/or ELCC staff knowledge on children undergoing oral immunotherapy or other food challenges were also not reported on. Nevertheless, common themes amongst the included studies were identified. We also recognize that articles from grey literature and studies outside Europe and North America and in languages other than English or French were excluded.

Our review also illustrates the need for further research on and understanding of effective and sustainable preventative and education strategies for ELCCs and service providers. As specific jurisdictions may adapt a diversity of policies and practices, our review can be utilized to inform food allergy and anaphylaxis management training and food-handling requirements for both private and public ELCC facilities and programs. Adequate food allergy training and food-handling knowledge may serve value to all staff who care for children with and without food allergies. The provision of available EAIs, whether prescribed or stock, should be considered mandatory. Finally, we support the need for the standardization of both core outcomes in food allergies, which would facilitate a comparison of studies, as well as the standardization of a baseline for food allergy management and education.

## 5. Conclusions

In conclusion, the current management of food allergies in early childhood centres in Western nations where data were available, including food-allergy-related policies, training, and meal provision, is diverse (Box 1). We suggest that annual food-handling and food allergy training, standardized diet forms for jurisdictions that provide children with meals on-site, and having available emergency anaphylaxis plans, EAIs, and communication plans with parents be considered by independent organizations and governing jurisdictions. This may increase the ELCC staff knowledge of and abilities and self-confidence in preventing and managing food allergies safely and effectively in childcare centres.

Box 1Key messages.
Younger children are at an increased risk of food allergic reactions because of developmentally appropriate behaviours;Although very rare, fatal allergic reactions have been documented in early learning and childcare centres;Early childhood educators play an important role in managing and preventing food allergic reactions in childcare centres;Early childcare educators have variable training and experience with food allergies;Educational interventions, communication tools, and strategical guideline implementation may be beneficial for childcare centres and can engage all stakeholders, including parents of children with and without food allergies.


## Figures and Tables

**Figure 1 children-10-01175-f001:**
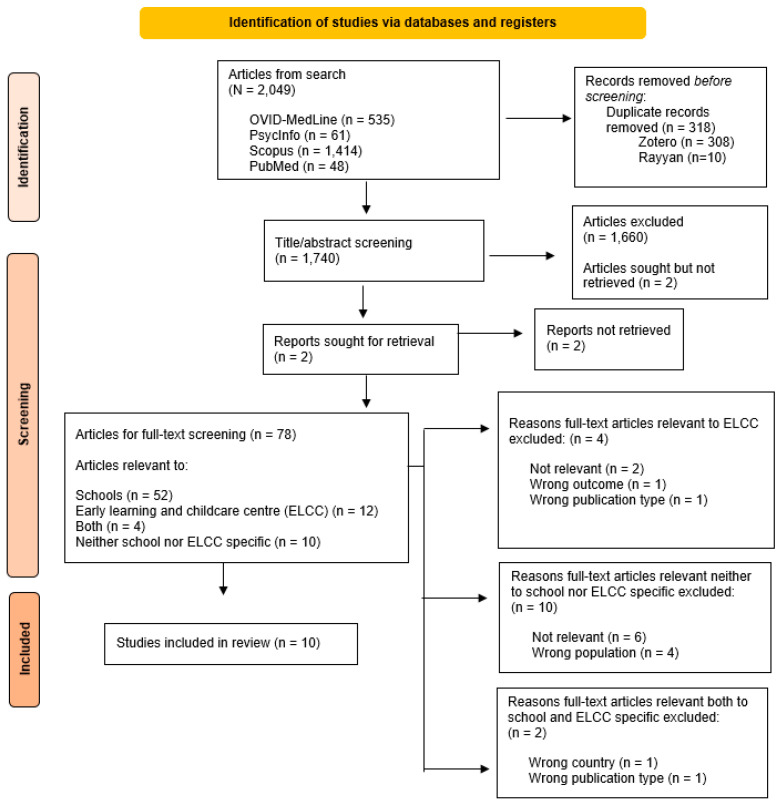
PRISMA flow diagram depicting the selection process for articles and reports relevant in the current scoping review.

**Table 1 children-10-01175-t001:** Summary of articles’ countries of origin, research designs, methods, and populations, presented in alphabetical order by first author’s last name.

First Author, Year, Country	Research Design	Methods	Site Description	Participants
Jacobsen 2018 Australia [31]	Descriptive cross-sectional	A 41-question online survey was sent to all ELCC directors in Western Australia. The questionnaire included both multiple-choice and open-ended questions. Data for the open-ended questions were collated and reported.	53 ELCCs Number of children enrolled across the sites (*n* = 4679) Number of children enrolled per site (*n* = 13–289; mean = 88)	Participants: (*n* = 53) Participant roles: Service directors (75%) Food coordinators (11%) Educators (6%) Owners (6%) Administration staff (2%)
Hua 2020 Australia [18]	Descriptive cross-sectional	Used Jacobsen et al. (2018) [31]’s online survey disseminated to registered members of the Australian Children’s Education and Care Quality Authority across all Australian jurisdictions.	Number of services not reported Number of children enrolled per site (*n* = 12–1000 children; median = 80)	Participants: (*n* = 494) Participant roles: Centre directors (*n* = 407; 82%) Educators (*n* = 22; 4.4%) Food service staff (*n* = 16; 3.2%) Owners (*n* = 10; 2%) Administrators (*n* = 14; 2.8%) Licencees (*n* = 10; 2%) Other staff members (*n* = 15; 3%) *
Erkkola 2015 Finland [33]	Quasi experimental pre- and post-intervention	As part of the “Promoting nutrition and allergy health in preschoolers” study and the Finnish Allergy Program, new special diet forms were implemented over three years. Data were collected through a survey (2013), site visit and questionnaire (2014), and kitchen staff phone interviews (2015). The prevalence of allergy diets across the study period, evaluation of foods avoided related to the intervention, and differences in numbers of avoided foods were evaluated statistically.	40 ELCCs Total number of children enrolled within participating sites: 2013 (*n* = 3216) 2014 (*n* = 3233) 2015 (*n* = 3411)	Questionnaire participants: Site personnel, including directors and kitchen staff (*n* = 414)
Kilger 2015 Germany [34]	Epidemiological cross-sectional	Three versions of a 22-item questionnaire were administered to schoolteachers, childcare providers, and parents in one German city. Questions about “emergency kits” referred to the kit contents, handling of the kit, and EAP. Parents’ responses were reported separately than those of teachers and childcare providers.	86 primary schools and ELCCs ** Number of children included in the study (*n* = 5981) Data specific to preschool-aged-children were not reported.	Participants: Parents (*n* = 5981) Childcare providers (*n* = 259) Schoolteachers (*n* = 112)
Dumeier 2018 Germany [32]	Quasi experimental pre- and post-intervention	Preschool teachers/ ECEs were invited to food allergy education sessions. Pre- and post-session questionnaires were administered pre- and post-session and at 4–12-weeks post-intervention. Post-intervention and 4–12-week post-intervention data were compared by the McNemar or Wilcoxon test. Bonferroni correction was applied, and the *p*-value was adjusted to *p* ≤ 0.025 to be significant.	39 ELCCs	Participants: Education session and pre-intervention questionnaire (*n* = 154) Practical EAI performance and post-intervention questionnaire (*n* = 125) 4–12-week follow-up assessment (*n* = 75)
MacGiobuin 2017 Ireland [35]	Cross-sectional	A 32-question online survey was administered to “Early years service providers”. Early years services included preschool, Montessori, and childcare centres. Some questions allowed for free-text answers.	Number of services not reported Number of children represented by services (*n* = 3203) Median number of children per site (*n* = 23, range = 2–128)	Participants: Early years service providers (*n* = 98)
Otten 2015 USA [38]	Qualitative	Early care and education providers who serve at least one meal per day to children 2–6 years old participated in a semi-structured interviewed. The interview consisted of 30 open-ended questions and a site tour. Research staff took still photos of the food preparation, storage, and service environment. Data were professionally transcribed verbatim and an inductive approach and ecological framework were used for analysis.	16 ELCCs (*n* = 12) Family home (*n* = 4) Site type: Single site (*n* = 9) Multi-site (*n* = 7) Mean number of children represented: 0–24 months (*n* = 21, range 3–5) 2–5 y (*n* = 39, range 4–98) 6+ y (*n* = 11, range 1–30)	Participants: Site director/owner (*n* = 14) Food service staff (*n* = 2)
Foster 2015 USA [36]	Quasi experimental pre- and post-intervention	ELCC directors, teachers/ECEs, and other staff participated in an anaphylaxis management educational session. Questionnaires were administered pre- and post-sessions. Baseline and post-intervention data were compared using Spearman rank correlation coefficients and the Kruskal–Wallis, Wilcoxon rank sum, χ^2^, and Fisher exact tests.	10 ELCCs Number of children enrolled across the sites (*n* = 168)	Participants: Pre-intervention survey (*n* = 181) Intervention and post-intervention survey (*n* = 171)
Lanser 2016 USA [37]	Quasi experimental pre- and post-intervention	An online needs assessment survey was administered to licenced ELCCs (excluding home-based centres). An educational session was provided by pediatric allergists. A pre- and post-test were administered to the session participants. Pooled data were compared at baseline and post-intervention. No participants responded for the follow-up test 6-months post-session.	72 ELCCs All centre-based Sizes of ELCCs: <20 children (*n* = 10) 20–50 children (*n* = 12) 50–80 children (*n* = 26) >80 children (*n* = 44)	Participants: Needs assessment survey (*n* = 93) Training curriculum (*n* = 45)
Wahl 2015 USA [39]	Quasi experimental pre- and post-intervention	School teachers, childcare providers, and camp staff were invited to participate in a training program with an education session. An initial survey was administered post-intervention, and a second online survey at 3–12-months post-intervention. Responses were grouped according to the time-interval post-intervention the data were collected (0–3 months; 3–6 months; 6–12 months; and >12 months). Participants who experienced a food allergy emergency post-intervention were followed up with a phone interview.	Number of participating sites not reported.	Participants: Primary survey (*n* = 4088) Secondary survey (*n* = 332) Phone interview (*n* = 53) Roles of participants: Teachers (48%) School Aides (5%) Administrators (5%) Childcare providers (6%) School Nurses (2%) Other (34%) ***

* Other staff member roles not reported. ** Authors referred to childcare centres as “kindergarten”, which enrolled children aged 1–5 years. *** Others include camp counsellors, bus drivers, multiples of specified job titles, parents, volunteers, coaches, food service workers, or no indication of job title. Abbreviations: EAI: epinephrine auto-injector; ECE: early childhood educator; ELCC: early learning and childcare centre; NS: not specified; USA: United States of America; y: years.

**Table 2 children-10-01175-t002:** Summary of policies, emergency action plans (EAPs), epinephrine auto-injector (EAI) availability, interventions, educational interventions, and other management practices among schools, presented in alphabetical order by first author’s last name.

First Author, Year	Protocols or Management Practices *	Food Allergy Training	Food Preparation and Provision	Intervention and Main Outcomes
Jacobsen 2018 [31]	A total of 96% of participating ELCCs had action plans per Western Australia government laws, and 96% of sites reported the staff knew where the EAIs were, 36% of which were in locked locations. A total of 68% of sites had stock EAIs available; 31% had more than one (range = 1–8 per site). A total of 58% of sites had an EAI training device.	A total of 91% of sites required all staff members (or all educators) to complete anaphylaxis management training, including EAI use, and 33% of food coordinators were also trained. A total of 7% of participating sites did not require training in first aid or anaphylaxis management.	Not reported	No intervention provided
Hua 2020 [18]	A total of 91% of participating ELCC staff reported their site required staff training on anaphylaxis management. More services in Victoria required training on recognizing allergic reactions, including anaphylaxis, compared to services based in Queensland and New South Wales (87% vs. 75% vs. 66%, respectively; χ^2^(2, *n* = 441) = 17.24, *p* < 0.001). A total of 97% of participants knew the location of the EAIs, although only 95.7% had access; 37% of EAIs were stored in locked locations. A total of 47.6% of participants reported their site had at least one stock EAI.	First-aid training, which included anaphylaxis management, was provided in 90.5% of services (*n* = 447/494), while 6.7% of services (*n* = 33/494) required no training.	Not reported	No intervention provided
Erkkola 2015 [33]	Children with health-based-diet requests were asked for a physician-signed medical certificate. The new form subjects the parents of children at risk of anaphylaxis to inform ELCC staff on how to use the EAI. Those with diagnosed food allergies were recommended that a physician follow up every 1–2 years. No further information on EAIs or EAPs were provided.	Not reported	A total of 75% of ELCCs received meals from a central distribution kitchen, and 25% of ELCCs prepared meals on-site.	The special diet form was gradually implemented in three cities. Centres still using old special diet forms and those with no forms were included in the control group analysis. The prevalence of allergy diets amongst preschoolers significantly decreased from 7.6% to 7.3% to 4.3% in 2013, 2014, and 2015, respectively (*p* < 0.001). There was also a decrease in the numbers of foods avoided per child at the 2015 follow-up.
Kilger 2015 [34]	A total of 53 parents reported that an “emergency kit” (including an EAP and medication) was prescribed for their child. The majority of medications in the kits were corticosteroids (69.8%), antihistamines (62.3%), and B2-antagonists (37.7%). Only 26.4% of children had an EAI prescribed.	Not reported	Not reported	No intervention provided
Dumeier 2018 [32]	Among ECEs who had children with food allergies under their care, 71% (43/61) reported that rescue medication was available on their sites.	Not reported	Not reported	A 1 h education session was provided. Session topics included food allergy epidemiology and mechanisms and allergy types and treatment. Practical EAI administration demonstration and evaluation of practical performance using training devices and dolls were included. Handouts were also provided. Post-intervention, more participants knew all four common anaphylactic triggers provided by authors (apples, chicken eggs, cow’s milk, peanuts, pollen, cat hair, insect sting). Perceived emergency preparedness increased from pre- to post-intervention from 11% (8/75) to 88% (66/75) (*p* < 0.001), and decreased to 79% at the 4–12-week follow-up (59/75, *p* < 0.001). Better EAI administration skills were seen post-intervention. Three out of eight drug administration problems were “high clinical risk”, including the injector was not unlocked, the injector was not activated, and the user’s finger was injected instead of the receiver. Confidence in EAI administration increased from pre- to- post-intervention and to follow-up (5/10 vs. 9/10 vs. 8/10, respectively; *p* < 0.001).
MacGiobuin 2017 [35]	A total of 74% of sites had EAIs, 42% had salbutamol inhalers, and 48% had antihistamines. Staff and parents checked the medications’ expiry dates. A total of 46 sites had written emergency plans, and 52% had individual EAPs available for food-allergic children. EAPs were written by staff or parents. None were signed by a physician.	A total of 19% had previous food allergy and/or anaphylaxis training, while 19% reported never having been trained. A total of 57/98 sites did not respond (i.e., missing data).	Not reported	No intervention provided
Otten 2015 [38]	Two photos of participating ELCCs showed detailed food allergy lists in food service areas used to manage and prepare for food allergy meals on-site.	Not reported	All sites provided meals to enrolled children. Six photo examples of kitchen and storage facilities were provided. Factors affecting food purchasing included jurisdictional law and subsidy programs, foodservice areas and equipment, parents driving food choices, and the individual needs of the children, including food allergies and accommodating cultural preferences.	No intervention provided
Lanser 2016 [37]	A total of 50.5% of participants worked in “peanut-free” ELCCs. EAPs were not available in 27% of participants’ workplaces.	A total of 43% of participants had previous formal or informal food-allergy-related training. A total of 36% of participants previously experienced a food allergic reaction at work.		A physician-led 1 h presentation was provided. Session topics included signs and symptoms of allergic reactions, treatment, food label laws, allergen avoidance, and use of EAP and EAI administration demonstration. Understanding of food allergies increased from pre- to post-intervention (62.6% vs. 85.2%, respectively; *p* < 0.001). Allergic reaction recognition increased from pre- to post-intervention (62.2% vs. 81.1%, respectively; *p* < 0.001). Knowledge of allergic reaction treatment increased from pre- to post-intervention (51.1% vs. 80.6%, respectively; *p* = 0.001). Food-labelling-law knowledge also increased from pre- to post-intervention (40.0% vs. 83.0%, respectively; *p* < 0.001). Overall, knowledge of all question categories increased from pre- to post-intervention (from 54.0% to 83.0%, respectively; *p* < 0.001).
Foster 2015 [36]	Not reported	A total of 70% of ELCC staff, including ECEs, had previous EAI training, and 57% had previous anaphylaxis recognition training.	Not reported	A physician-led 40 min education session was provided. Session topics included anaphylaxis recognition and treatment, EAI demonstration, and practice session with a training device. From pre- to post-intervention, there was increased knowledge on correct steps for EAI administration (3/5 vs. 4.2/5, respectively; *p* < 0.001), comfort level with anaphylaxis symptom recognition (5.1/10 vs. 8.7/10, respectively; *p* < 0.001), and comfort level with EAI administration (5.4/10 vs. 8.8/10, respectively; *p* < 0.001). A total of 99% of participants believed the session had been beneficial, and 76% wanted annual training.
Wahl 2015 [39]	Not reported	A total of 42% (*n* = 105) of ECEs reported having previous food allergy training.	Not reported	A nurse-led 45 min education session was provided. Session topics included food allergy facts, allergic reaction prevention, symptom recognition, and treatment. EAI administration demonstration and practical training were also provided. Amongst all participants, including ECEs, >90% reported feeling more confident in preventing, recognizing, and knowing what to do in cases of emergency. Amongst responders at the 3–12-month follow-up, >90% responded their sustained confidence. A total of 94% of participants reported being confident in administering an EAI post-intervention. Post-intervention, 62% reported their food allergy management practices were changing. A total of 21 participants, one of whom was an ECE, participated in a food-allergy-emergency post-intervention. Of these participants, 62% reported that the educational intervention was useful and 52% reported that the hands-on EAI training was useful in responding to actual emergencies.

* Including EAP and EAI availability. Abbreviations: EAI: epinephrine auto-injector; ELCC: early learning and childcare centre; EAP: emergency allergy plan; ECE: early childhood educator; USA: United States of America; y: year.

## Data Availability

Not applicable; all publications included in this scoping review are available on academic databases.

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
