# Peer review of "Food Allergy Education and Management in Early Learning and Childcare Centres: A Scoping Review on Current Practices and Gaps"

_children, 2023, doi:10.3390/children10071175_

Round 1

Reviewer 1 Report

Very interesting review contribution. Brings useful information and identifies issues to be addressed. Very well described paper selection process, justified references' choice.

Not much to be reproached to the authors. I do not feel qualidfied to assess English language quality.

Reviewer 2 Report

Thanks for asking me to review this scoping review of current practices and gaps in food allergy education and management in early learning and childhood centres.

I think the review was quite thoroughly done, however it was a little difficult to read in parts. My comments are intended to help with the readablility of this paper.

1. The tables might be easier to read if they were in landscape

2. Page 9. Could you begin the paragraph on training by discussing the difference types of training that are required, and then begin the discussion about the level of training received by the child care educators surveyed?

3. page 15. Think about your use of abbreviations. There are so many abbreviations that look similar, particularly in the last half of this page I found the text difficult to read.

4. Discussion - I would argue that most pre-school children are pre-literate, . hence they are at specific risk of accidental exposure to their allergen/s.

no additional comments

Reviewer 3 Report

This manuscript is excellent in every sense.  It covers a subject that has been largely neglected but yet is very important.  The manuscript was comprehensive, well organized, and well written.

On line 407, I did not that there are two words that need to be separated by a space.
